# An ECF-type transporter scavenges heme to overcome iron-limitation in *Staphylococcus lugdunensis*

**Angelika Jochim[1], Lea Adolf[1], Darya Belikova[1], Nadine Anna Schilling[2], Inda Setyawati[3], Denny Chin[4], Severien Meyers[5], Peter Verhamme[5], David E Heinrichs[4], Dirk J Slotboom[3], Simon Heilbronner[1,6,7]***

[1]Interfaculty Institute of Microbiology and Infection Medicine, Department of Infection Biology, University of Tübingen, Tübingen, Germany; [2]Institute of Organic Chemistry, University of Tübingen, Tübingen, Germany; [3]Groningen Biomolecular Sciences and Biotechnology Institute, University of Groningen, Groningen, Netherlands; [4]Department of Microbiology and Immunology, University of Western Ontario, London, Canada; [5]Center for Molecular and Vascular Biology, Leuven, Belgium; [6]German Centre for Infection Research (DZIF), Partner Site Tübingen, Tübingen, Germany; [7](DFG) Cluster of Excellence EXC 2124 Controlling Microbes to Fight Infections, Tübingen, Germany

**Abstract** Energy-coupling factor type transporters (ECF) represent trace nutrient acquisition systems. Substrate binding components of ECF-transporters are membrane proteins with extraordinary affinity, allowing them to scavenge trace amounts of ligand. A number of molecules have been described as substrates of ECF-transporters, but an involvement in iron-acquisition is unknown. Host-induced iron limitation during infection represents an effective mechanism to limit bacterial proliferation. We identified the iron-regulated ECF-transporter Lha in the opportunistic bacterial pathogen *Staphylococcus lugdunensis* and show that the transporter is specific for heme. The recombinant substrate-specific subunit LhaS accepted heme from diverse host-derived hemoproteins. Using isogenic mutants and recombinant expression of Lha, we demonstrate that its function is independent of the canonical heme acquisition system Isd and allows proliferation on human cells as sources of nutrient iron. Our findings reveal a unique strategy of nutritional heme acquisition and provide the first example of an ECF-transporter involved in overcoming host-induced nutritional limitation.

**\*For correspondence:**
simon.heilbronner@uni-tuebingen.de

**Competing interests:** The authors declare that no competing interests exist.

## Introduction

Trace nutrients such as metal ions and vitamins are needed as prosthetic groups or cofactors in anabolic and catabolic processes and are therefore crucial for maintaining an active metabolism. Metal ions such as iron, manganese, copper, zinc, nickel and cobalt must be acquired from the environment by all living organisms. In contrast many prokaryotes are prototrophic for vitamins like riboflavin, biotin and vitamin $B_{12}$. However, these biosynthetic pathways are energetically costly (*Roth et al., 1993*), and prokaryotes have developed several strategies to acquire these nutrients from the environment. ABC transporters of the Energy-coupling factor type (ECF-transporters) represent highly effective trace nutrient acquisition systems (*Erkens et al., 2012*; *Finkenwirth and Eitinger, 2019*). In contrast to the substrate-binding lipoproteins/periplasmic proteins of conventional ABC transporters, the specificity subunits of ECF transporters (ECF-S) are highly hydrophobic membrane proteins (6–7 membrane spanning helices) (*Erkens et al., 2012*). ECF-S subunits display a remarkably high affinity for their cognate substrates in the picomolar to the low nanomolar range,

which allows scavenging of smallest traces of their substrates from the environment (*Rempel et al., 2019*). Whether ECF type transporters can be used to acquire iron or iron-containing compounds is unknown.

The dependency of bacteria on trace nutrients is exploited by the immune system to limit bacterial proliferation by actively depleting nutrients from body fluids and tissues. This strategy is referred to as 'nutritional immunity' (*Hood and Skaar, 2012*; *Cassat and Skaar, 2013*). In this regard, depletion of nutritional iron ($Fe^{2+}$/$Fe^{3+}$) is crucial as iron is engaged in several metabolic processes such as DNA replication, glycolysis, and respiration (*Schaible and Kaufmann, 2004*; *Weinberg, 2000*). Extracellular iron ions are bound by high-affinity iron-chelating proteins such as lactoferrin and transferrin found in lymph and mucosal secretions and in serum, respectively. However, heme is a rich iron source in the human body and invasive pathogens can access this heme pool by secreting hemolytic factors to release hemoglobin or other hemoproteins from erythrocytes or other host cells. Bacterial receptors then extract heme from the hemoproteins. This is followed by import and degradation of heme to release the nutritional iron. To date, several heme acquisition systems of different Gram-positive and Gram-negative pathogens have been characterized at the molecular level (see *Choby and Skaar, 2016* for an excellent review).

Staphylococci are a major cause of healthcare-associated infections that can lead to morbidity and mortality. The coagulase-positive *Staphylococcus aureus* represents the best-studied and most invasive species. Coagulase-negative staphylococci (CoNS) are regarded as less pathogenic than *S. aureus* and infections caused by CoNS are normally subacute and less severe. In this regard, the CoNS *Staphylococcus lugdunensis* represents an exception. *S. lugdunensis* infections frequently show a fulminant and aggressive course of disease that resembles that of *S. aureus*. Strikingly, *S. lugdunensis* is associated with a series of cases of infectious endocarditis (*Liu et al., 2010*). The reasons for the apparently high virulence potential of *S. lugdunensis* remain largely elusive and few virulence factors have been identified so far. In this respect, it is interesting to observe that *S. lugdunensis*, unlike other staphylococci but similar to *S. aureus*, encodes an iron-dependent surface determinant locus (Isd) system (*Heilbronner et al., 2011*; *Heilbronner et al., 2016*). Isd facilitates the acquisition of heme from hemoglobin and can be regarded as a hallmark of adaption towards an invasive lifestyle. However, to ensure continuous iron acquisition within the host, many pathogens encode multiple systems to broaden the range of iron-containing molecules that can be acquired (*Sheldon et al., 2016*).

Here we report the identification of an iron-regulated ECF-type ABC transporter (named LhaSTA) in *S. lugdunensis*. We found LhaSTA to be specific for heme, thus representing a novel strategy to overcome nutritional iron limitation. Recombinant LhaS accepted heme from several host hemoproteins such as hemoglobin, myoglobin or hemopexin. Consistent with these data, LhaSTA expression allowed proliferation of *S. lugdunensis* in the presence of these iron sources as well as human erythrocytes or cardiac myocytes as a sole source of nutrient iron. Our data indicate that LhaSTA function is independent of the presence of surface-displayed hemoprotein receptors suggesting Isd-independent acquisition of heme from host hemoproteins. Our work identifies LhaSTA as the first ECF transporter that facilitates iron acquisition, thus participating to overcome host immune defenses.

## Results

### LhaSTA encodes an iron regulated ECF transporter

The *isd* locus of *S. lugdunensis* shows several characteristics that distinguish it from the locus of *S. aureus*. Amongst these is the presence of three genes that encode a putative ABC transporter and are located between *isdJ* and *isdB* (*Figure 1A*; *Heilbronner et al., 2011*). Analysis of the open reading frame using Pfam (*El-Gebali et al., 2019*) revealed that the three adjacent genes encode components of a putative ECF-transporter, namely a specificity subunit (*lhaS* - SLUG_00900), a transmembrane subunit (*lhaT* - SLUG_00910) and an ATPase (*lhaA* - SLUG_00920), and they might be part of a polycistronic transcript. The location within the *isd* operon suggested a role of the transporter in iron acquisition. Bacteria sense iron limitation using the ferric uptake repressor (Fur) which forms DNA-binding dimers in the presence of iron ions (*Coy and Neilands, 1991*). Under iron limitation, Fe dissociates from Fur and the repressor loses affinity for its consensus sequence (*fur* box) allowing transcription. Interestingly, a *fur*-box was located upstream of *lhaS* (*Figure 1A*). qPCR

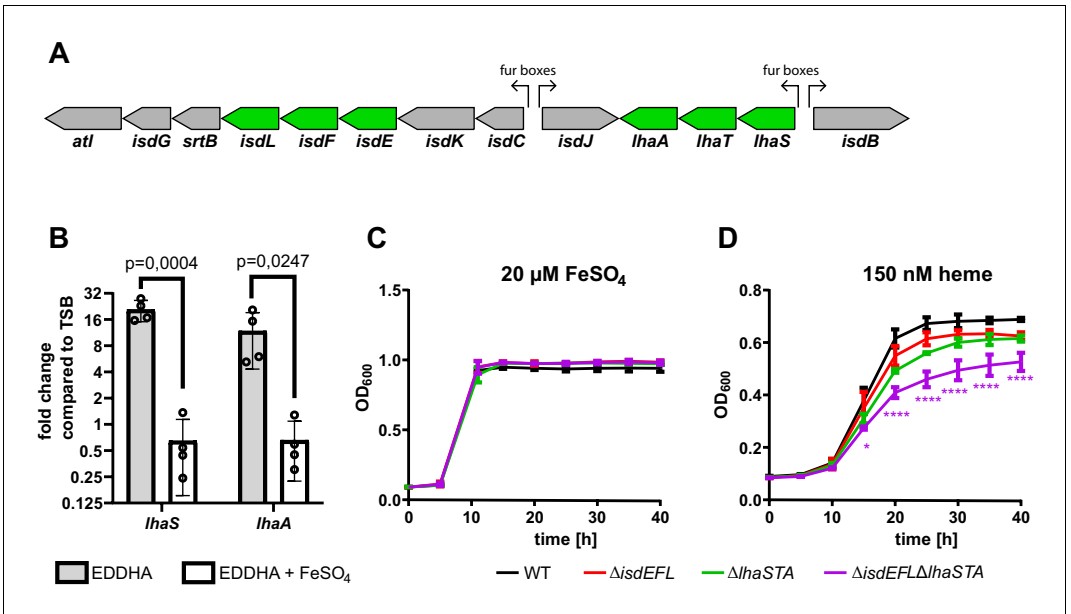

**Figure 1.** LhaSTA represents an iron-regulated heme transporter. (A) Schematic diagram of the *isd* operon of *S. lugdunensis* N920143. Coding sequences, direction of transcription and Fur-binding sites are indicated. ABC membrane-transporters are shown in green. *lhaS* - SLUG_00900; *lhaT* - SLUG_00910; *lhaA* - SLUG_00920 (B) Iron-regulated expression of Lha: *S. lugdunensis* was grown overnight in TSB, TSB + 200 µM EDDHA or TSB + 200 µM EDDHA + 200 µM FeSO$_4$. Gene expression was quantified by qPCR. Expression was normalized to 5srRNA and to the TSB standard condition using the ΔΔCt method. Fold differences in gene expression are shown. Data represent mean and SD of four independent experiments. Statistical evaluation was performed using students unpaired t-test (lhaS: t = 7,045, df = 6; lhaA: t = 2,979, df = 6) C/D Growth curves of *S. lugdunensis* N920143 and isogenic mutants. The wild type (WT) *S. lugdunensis* N920143 strain and the indicated isogenic null mutant strains were grown in the presence of 20 µM FeSO$_4$ (C) or 150 nM heme (D) as a sole source of iron. 500 µl of bacterial cultures were inoculated to an OD$_{600}$ = 0,05 in 48 well plates and OD$_{600}$ was monitored every 15 min using an Epoch1 plate reader. For reasons of clarity values taken every 5 hr are displayed. Mean and SD of three experiments are shown. Statistical analysis was performed using one-way ANOVA followed by Dunett's test for multiple comparisons. * - p<0,05, ****p<0,00001.

analysis in *S. lugdunensis* N920143 revealed that the expression of *lhaS* and *lhaA* increased ~21 and~12 fold, respectively, in the presence of the Fe-specific chelator EDDHA (*Figure 1B*). The effect of EDDHA could be prevented by addition of FeSO$_4$ (*Figure 1B*). This confirmed iron-dependent regulation and suggested that LhaSTA is involved in iron acquisition.

## LhaSTA allows bacterial proliferation on heme as a source of nutrient iron

LhaSTA is encoded within the *isd* operon and the Isd system facilitates the acquisition of heme from hemoglobin (*Heilbronner et al., 2016*; *Zapotoczna et al., 2012*). Therefore, we speculated that LhaSTA might also be involved in the transport of heme. To test this, we used allelic replacement and created isogenic deletion mutants in *S. lugdunensis* N920143 lacking either *lhaSTA* or *isdEFL*, the latter of which encodes the conventional lipoprotein-dependent heme transporter of the *isd* locus. Further, we created a Δ*lhaSTA*Δ*isdEFL* double mutant. In the presence of 20 µM FeSO$_4$ all strains showed similar growth characteristics (*Figure 1C*). The two single mutants had a slight growth defect compared to wild type when heme was the only iron source. However, the Δ*lhaSTA*Δ*isdEFL* mutant showed a significant growth defect under these conditions (*Figure 1D*). These data strengthen the hypothesis that LhaSTA is a heme transporter.

## LhaS binds heme

To confirm the specificity of LhaSTA, we heterologously produced the substrate-specific component LhaS in *E. coli* and purified the protein. We observed that the recombinant protein showed a distinct

red color when purified from *E. coli* grown in rich LB medium, which contains heme due to the presence of crude yeast extract (*Fyrestam and Östman, 2017*; *Figure 2A*). The absorption spectrum of the protein showed a Soret peak at 415 nm and Q-band maxima at 537 and 568 nm, suggesting histidine coordination of the heme group. Both the visible color and the spectral peaks were absent when LhaS was purified from *E. coli* grown in heme-deficient RPMI medium (*Figure 2A*). We conducted MALDI-TOF analysis of holo-LhaS purified from *E. coli* grown in LB and identified two peaks, one of which corresponds to full length recombinant LhaS (24074.437 Da expected weight), and the other to heme (616,1767 Da expected weight). Importantly, the heme peak was not detectable when LhaS was purified from RPMI (*Figure 2B and C*). Furthermore, ESI-MS confirmed the presence of heme only in LhaS samples purified from LB (*Figure 2—figure supplement 1*). Using the extinction coefficients of LhaS and heme we calculated a heme-LhaS binding stoichiometry of 1:0.6 for the complex isolated from heme containing medium.

## LhaSTA represents an iron acquisition system

Next, we sought to investigate whether LhaSTA represents a functional and autonomous iron acquisition system. LhaSTA is located within the *isd* locus of *S. lugdunensis,* which, besides the conventional heme membrane transporter IsdEFL, also encodes the hemoglobin receptor IsdB and the cell wall-anchored proteins IsdJ and IsdC. Furthermore, the locus encodes the putative secreted/membrane-associated hemophore IsdK, whose role in heme binding or transport is currently unknown (*Zapotoczna et al., 2012*), and the autolysin *atl* remodeling the cell wall (*Farrand et al., 2015*). To study solely LhaSTA-dependent effects, we disabled all known heme import activity in *S. lugdunensis* by creation of a deletion mutant lacking the entire *isd* operon (from the *atl* gene to *isdB*, *Figure 1A*). Then we expressed *lhaSTA* under the control of its native promoter on a recombinant plasmid in the Δ*isd* background. *S. lugdunensis* has been reported to degrade nutritional heme in an IsdG-independent fashion due to an unknown enzyme (OrfX) (*Haley et al., 2011*). Therefore, we speculated that heme degradation in this strain might still be possible. LhaSTA deficient and proficient strains showed comparable growth in the presence of $FeSO_4$ (*Figure 3—figure supplement 1*). However, only the *lhaSTA* expressing strain was able to grow in the presence of heme as sole source of nutrient iron (*Figure 3A*). To further support a role for LhaSTA in iron import, we isolated the cytosolic fraction of the strains prior and after incubation with heme and measured iron levels using the ferrozine assay (*Riemer et al., 2004*). Consistently, we found that LhaSTA expression increased cytosolic iron levels post incubation with heme (*Figure 3B*). These data suggest that LhaSTA represents a 'bona fide' and functional autonomous iron acquisition system.

## LhaSTA enables acquisition of heme from various host hemoproteins

We wondered how *S. lugdunensis* might benefit from a heme specific ECF-transporter when a heme acquisition system is already encoded by the canonical Isd system. Indeed, Isd represents a highly effective heme acquisition system. Interactions between the surface receptor IsdB and the proteinaceous part of hemoglobin are thought to enhance heme release to increase its availability (*Bowden et al., 2018*; *Gianquinto et al., 2019*). The downside of this mechanism is the specificity for hemoglobin because heme derived from other host hemoproteins such as myoglobin remains inaccessible. In contrast, the HasA hemophore produced by Gram negative pathogens is reported to bind heme with sufficient affinity to enable heme acquisition from a range of host hemoproteins without the need of protein-protein interactions to enhance heme release (*Deniau et al., 2003*; *Wandersman and Delepelaire, 2004*). As ECF-transporters are known to have high affinity towards their ligands, we speculated that LhaS might represent a membrane-located high affinity 'hemophore' allowing heme acquisition from hemoproteins other than hemoglobin. We explored this idea using the hemoprotein myoglobin which is abundant in muscle tissues. Myoglobin was previously reported not to interact with *S. lugdunensis* IsdB (*Zapotoczna et al., 2012*) and is therefore unlikely to be a substrate for the Isd system. We analyzed the growth of the *S. lugdunensis* wild type (WT) strain as well as of the isogenic *lhaSTA* deficient strain (*Figure 4A*) on human hemoglobin (hHb) or on equine myoglobin (eqMb) as sole sources of nutrient iron (*Figure 4B*). Unlike the WT strain, the *lhaSTA* deficient strain displayed a mild proliferation defect on hHb and a pronounced growth defect on eqMb (*Figure 4B*). Interestingly, *lhaSTA* deficiency did not impact proliferation on hemoglobin-haptoglobin (Hb-Hap) complexes suggesting Isd-dependent acquisition of heme from Hb-Hap.

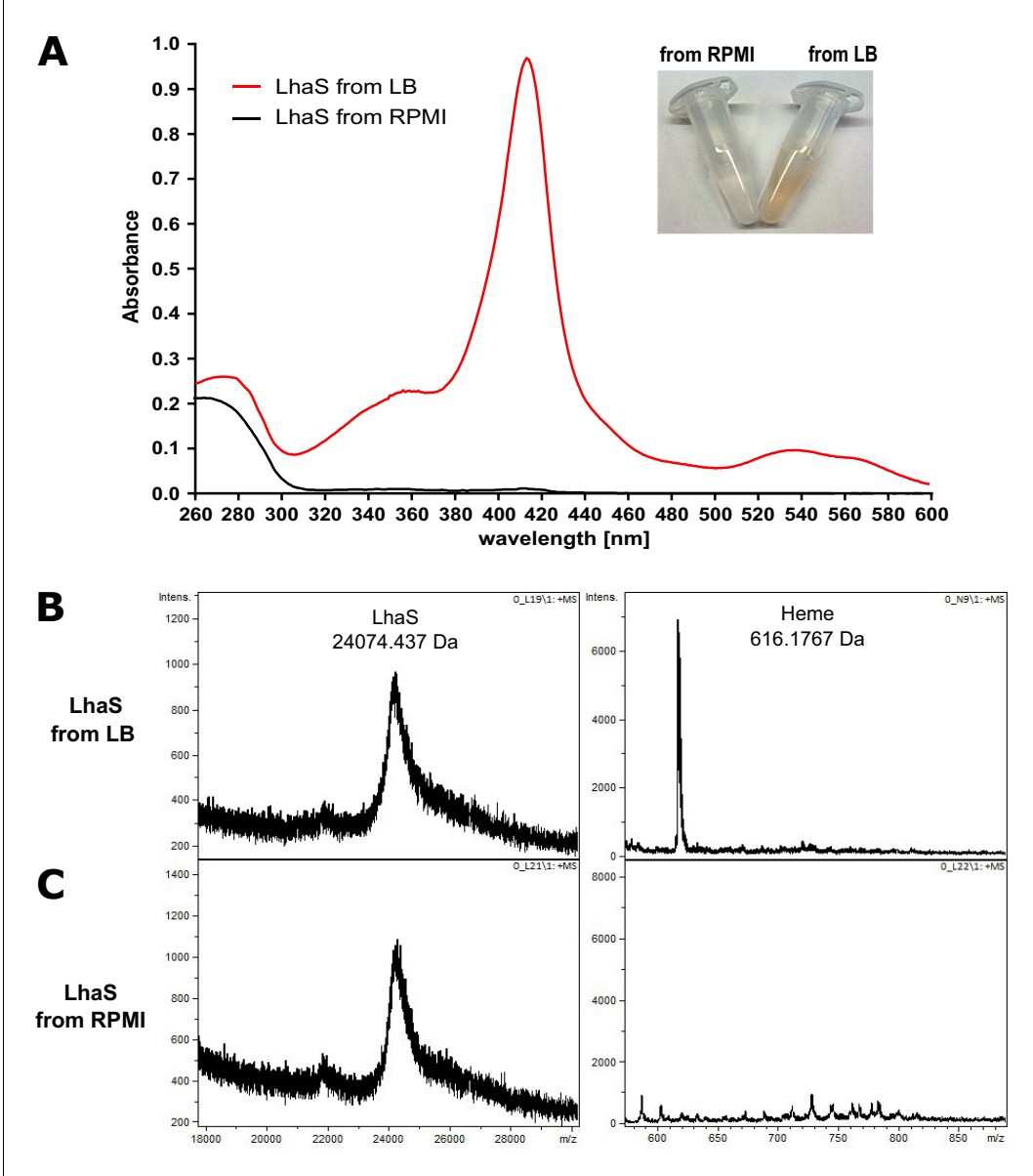

**Figure 2.** LhaS binds heme. (**A**) Ultraviolet-visible (UV-vis) spectrum of recombinant LhaS. C-terminal His-tagged LhaS was heterologously expressed in *E. coli* and purified from heme-containing LB medium or heme-free RPMI medium. The UV-vis spectrum of the purified LhaS was measured with a BioPhotometer. (**B**) and (**C**) MALDI-TOF mass spectra of recombinant LhaS. LhaS (**B**) was purified out of LB medium and apo-LhaS (**C**) was purified out of RPMI medium. Mass spectra were recorded with a Reflex IV in reflector mode. All spectra are a sum of 50 shots. Prior to measurements the protein samples were mixed with a 2,5-dihydroxybenzoic acid matrix dissolved in water/acetonitrile/trifluoroacetic acid (50/49.05/0.05) at a concentration of 10 mg ml$^{-1}$ and spotted onto the MALDI polished steel sample plate.

The online version of this article includes the following figure supplement(s) for figure 2:

**Figure supplement 1.** High resolution mass spectra of apo- and holo-LhaS.

These data strongly indicate that LhaSTA possesses a hemoprotein substrate range that differs from that of the Isd system. To further validate this, we used the above-described *S. lugdunensis isd* mutant expressing *lhaSTA* (***Figure 4A***) and tested its ability to proliferate on a range of different hemoproteins. (***Figure 4C***). We found that *lhaSTA* expression enabled growth on hemoglobin (human and murine origin) and myoglobin (human and equine origin) as well as with hemopexin

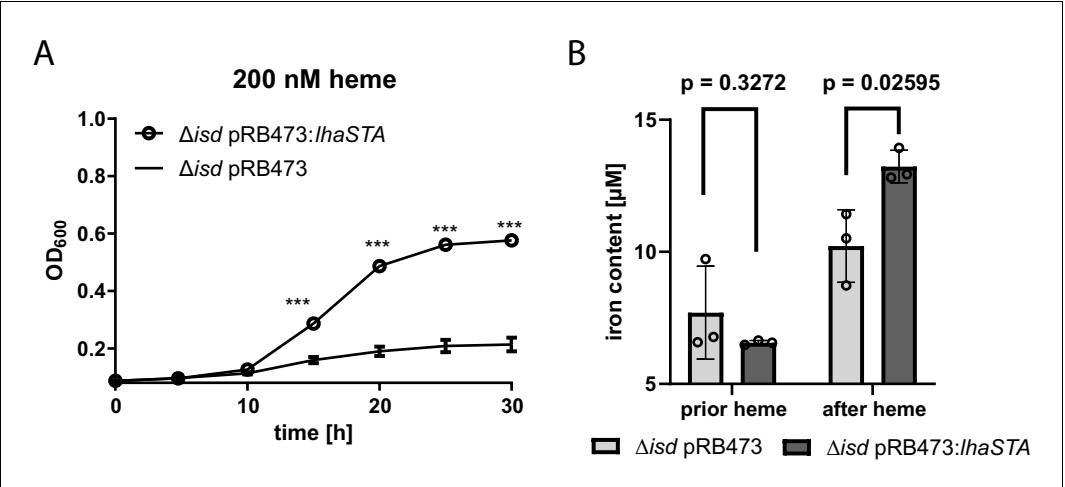

**Figure 3.** LhaSTA represents a functionally autonomous iron acquisition system. (**A**) LhaSTA-dependent proliferation. *S. lugdunensis* N920143 deletion mutant strains lacking the entire *isd* operon and expressing LhaSTA (Δ*isd* pRB473:*lhaSTA*) or not (Δ*isd* pRB473) from the plasmid pRB473 were grown in the presence of 200 nM heme as a sole source of iron. 500 µl of cultures were inoculated to an $OD_{600}$ = 0,05 in 48 well plates and $OD_{600}$ was monitored every 15 min using an Epoch1 plate reader. For reasons of clarity values taken every 5 hr are displayed. Mean and SD of three experiments are shown. Statistical analysis was performed using students unpaired t-test. ***p<0,0001 (**B**) Intracellular accumulation of iron. Strains were grown in iron limited medium to $OD_{600}$ = 0,6 and 5 µM heme were added for 3 hr. Cell fractionation of 1 ml $OD_{600}$ = 50 was performed and the iron content of the cytosolic fraction was determined using the ferrozine assay. Data represent the mean and SD of three independent experiments. Statistical analysis was performed using students unpaired t-test (t = 5,12729, df = 4). The online version of this article includes the following figure supplement(s) for figure 3:

**Figure supplement 1.** LhaSTA dependent growth.

(Hpx). Consistent with the above observations, Hb-Hap complexes did not enable growth of the *lhaSTA* proficient strain strengthening the notion that Hb-Hap acquisition is Isd-dependent. These data further indicate that LhaSTA allows extraction and usage of heme from a diverse set of host hemoproteins, thus expanding the range of hemoproteins accessible to *S. lugdunensis.*

To confirm the activity of LhaSTA at the biochemical level, we isolated *E. coli*-derived membrane vesicles that carried apo-LhaS. Following incubation of the vesicles with or without different host hemoproteins, LhaS was purified using affinity chromatography. Heme saturation of LhaS was assessed using SDS-PAGE and tetramethylbenzidine (TMBZ) staining, a reagent that turns green in the presence of hemin-generated peroxides (*Thomas et al., 1976*; *Figure 4D*). In the absence of hemoproteins during incubation, apo-LhaS did not stain with TMBZ, but TMBZ staining was observed after incubation with all the hemoproteins tested except for Hb-Hap. These data correlate with the ability of the *lhaSTA* proficient strain to grow on all hemoproteins but Hb-Hap complexes.

## LhaSTA allows usage of human host hemoproteins as an iron source

Usage of host derived hemoproteins requires the combined action of hemolytic factors to damage host cells as well as hemoprotein acquisition systems to use the released hemoproteins. Since we realized that the *S. lugdunensis* N920143 strain is non-hemolytic on sheep blood agar, we reproduced the Δ*isd* deletion as well as the plasmid-based expression of *lhaSTA* in the hemolytic *S. lugdunensis* N940135 strain. As for N920143, LhaSTA-dependent usage of hemoproteins was also observed in the N940135 background (*Figure 4—figure supplement 1*).

We speculated that the expression of LhaSTA is beneficial to *S. lugdunensis* during invasive disease as it allows usage of a wide range of hemoproteins as iron sources. To test this, we attempted to establish septic disease models for *S. lugdunensis*. However, we found that *S. lugdunensis* N940135 was unable to establish systemic disease in mice. Even when infected with $3*10^7$ CFU/animal, mice did not show signs of infection (weight loss/reduced movement). Three days post infection, the organs of infected animals showed low bacterial burdens frequently approaching sterility

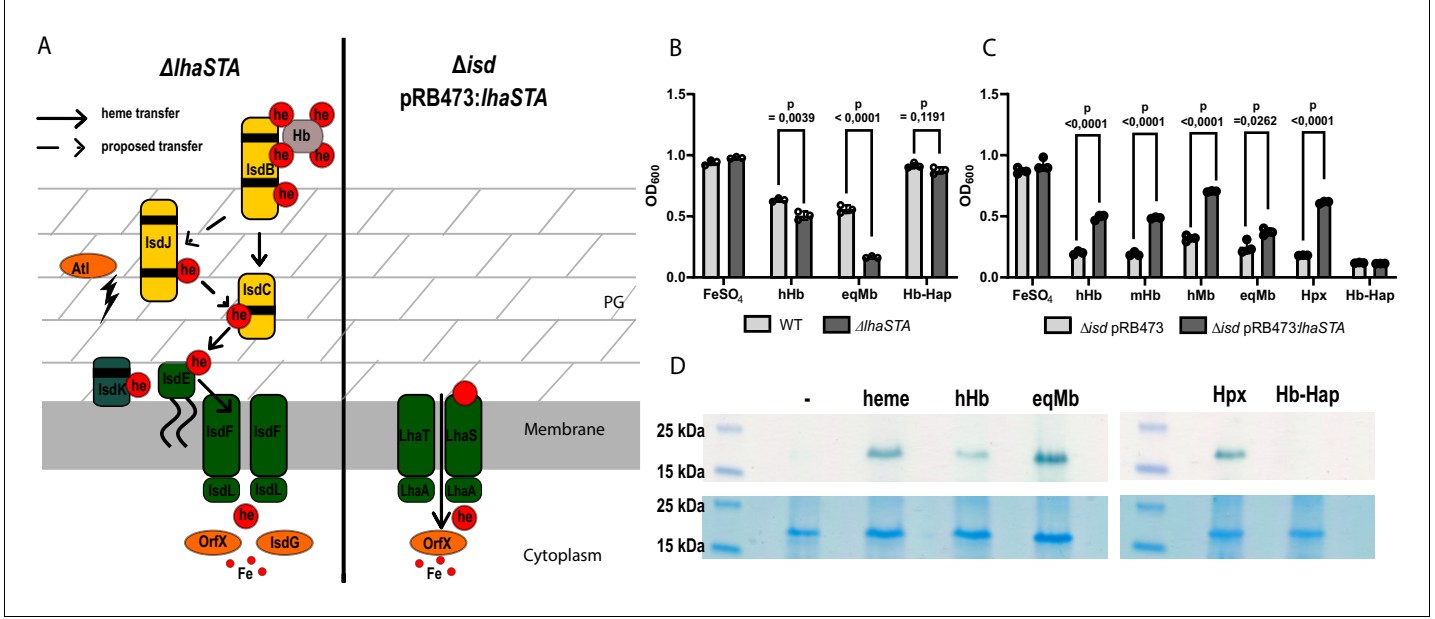

**Figure 4.** LhaSTA facilitates heme acquisition from a wide range of hemoprotein substrates. (A) Schematic diagram of known heme acquisition systems in the *S. lugdunensis* mutant strains lacking either the genes encoding LhaSTA (Δ*lhaSTA*, left) or the entire *isd* operon and expressing LhaSTA from the plasmid pRB473 (Δ*isd* pRB473:*lhaSTA*). ABC membrane transporters are shown in green. Cell wall-anchored proteins of the Isd-system are shown in yellow. Heme/hemoglobin-binding NEAT motifs within each protein are indicated as black boxes. Black arrows indicate the transfer of heme. he: heme; hb: hemoglobin; PG: peptidoglycan. (B) Growth of *S. lugdunensis* N920143 wild type (WT) and Δ*lhaSTA*. Strains were grown in the presence of 20 μM FeSO$_4$ or 2.5 μg/ml human hemoglobin (hHb) or 10 μg/ml equine myoglobin (eqMb) or 117 nM hemoglobin-haptoglobin complex (Hb-Hap) as a sole source of iron. 500 μl of cultures were inoculated to an OD$_{600}$ = 0,05 in 48 well plates and OD$_{600}$ was measured after 30 hr using an Epoch1 plate reader. Mean and SD of three experiments are shown. Statistical analysis was performed using students unpaired t-test. hHb - t = 6,0007, df = 4; eqMb – t = 20,52, df = 4; Hb-Hap – t = 1,978, df = 4. (C) Growth of *S. lugdunensis* N920143 Δ*isd* pRB473 and Δ*isd* pRB473:*lhaSTA*. Strains were grown in the presence of 20 μM FeSO$_4$ or 2.5 μg/ml hHb or 2.5 μg/ml murine hemoglobin (mHb) or 10 μg/ml human myoglobin (hMb) or 10 μg/ml eqMb or 200 nM human hemopexin (Hpx) or 117 nM Hb-Hap as a sole source of iron. 500 μl of cultures were inoculated to an OD$_{600}$ = 0,05 in 48 well plates and OD$_{600}$ was measured after 30 hr using an Epoch1 plate reader. Mean and SD of three experiments are shown. Statistical analysis was performed using students unpaired t-test hHb – t = 18,5, df = 4; mHb – t = 29,03, df = 4; hMb – t = 25,98, df = 4; eqMb – t = 3,442, df = 4; Hpx – t = 77,12 df = 4; Hb-Hap t = 2758 df = 4. (D) TMBZ-H$_2$O$_2$ stain of TGX gels for heme-associated peroxidase activity. Membrane vesicles were saturated with excess of hemoprotein (5.6 μM heme, 476 μg/ml hHb, 437 μg/ml eqMb, 5.6 μM Hpx, 476 μg/ml Hb-Hap) or no hemoprotein (-) for 10 min at RT. LhaS was purified, 15 μg protein was loaded on a TGX gel and stained for peroxidase activity with TMBZ-H$_2$O$_2$ (upper panel). Gels were destained and subsequently stained with BlueSafe (lower panel) to confirm the presence of the protein in all conditions.

The online version of this article includes the following figure supplement(s) for figure 4:

**Figure supplement 1.** Growth of *S. lugdunensis* N940135 Δ*isd* pRB473 and Δ*isd* pRB473:*lhaSTA*.

(*Figure 5—figure supplement 1*) and the expression of LhaSTA did not increase the bacterial loads within the organs. We speculate that the presence of human-specific but lack of mouse-specific virulence factors might reduce *S. lugdunensis* pathogenesis in mice. Little is known about virulence factors encoded by *S. lugdunensis*, however, human specific toxins that lyse erythrocytes to release nutritional hemoglobin have previously been described for *S. aureus* (*Spaan et al., 2015*). To further assess this, we performed hemolysis assays using human as well as murine erythrocytes (*Figure 5—figure supplement 2*). Hemolytic activity of *S. lugdunensis* culture filtrates was low compared to those of *S. aureus*. Nevertheless, we observed lysis of human erythrocytes while murine erythrocytes were not affected by *S. lugdunensis* culture filtrates. This suggests human specific factors mediating host cell damage (*Figure 5—figure supplement 2*).

Therefore, we used an ex-vivo model to investigate whether LhaSTA facilitates the usage of human cells as a source of iron. First, we supplied freshly isolated human erythrocytes as a source of hemoglobin. *Figure 5A* shows, that the presence of human erythrocytes significantly improved the growth of the Isd deficient but LhaSTA-expressing strain. Secondly, we used a human cardiac myocyte cell line as a source of iron. *S. lugdunensis* is associated with infective endocarditis and

myocytes are a source of myoglobin which can be acquired via LhaSTA. Indeed, we found that *lhaSTA* expression enhanced the growth of S. *lugdunensis* in the presence of cardiac myocytes.

In conclusion, our results suggest that LhaSTA represents a novel broad-range heme-acquisition system that expands the hemoprotein substrate range accessible to *S. lugdunensis* to overcome nutritional iron restriction (*Figure 6*).

## Discussion

Nutritional iron restriction represents an effective host strategy to prevent pathogen proliferation within sterile tissues. In turn, bacterial pathogens have developed a range of strategies to overcome nutritional iron limitation during infection. Amongst these is the production and acquisition of sidero-phores which scavenge the smallest traces of molecular iron to make it biologically available. The highly virulent *S. aureus* species produces the siderophores staphyloferrin A (SF-A) and staphylofer-rin B (SF-B) which are important during infection (*Beasley et al., 2011*; *Sheldon and Heinrichs, 2015*). *S. lugdunensis* is associated with a series of cases of infective endocarditis and the course of disease mimics that of *S. aureus* endocarditis. In contrast to *S. aureus*, *S. lugdunensis* does not pro-duce endogenous siderophores (*Brozyna et al., 2014*), suggesting that the iron requirements during infection must be satisfied through alternative strategies. Host hemoproteins can be used by patho-gens to acquire iron-containing heme and a plethora of hemoproteins are available during infection. Hemoglobin or myoglobin becomes available if the intracellular pool of the host is tapped by secre-tion of hemolytic factors. Alternatively, host hemopexin or hemoglobin-haptoglobin complexes involved in heme/hemoglobin turnover are extracellularly available to pathogens. Host hemoproteins are characterized by a remarkable affinity towards heme: Both, globin and hemopexin bind heme with dissociation constants (Kds) smaller than 1 pM (*Hargrove et al., 1996*; *Tolosano and Altruda,*

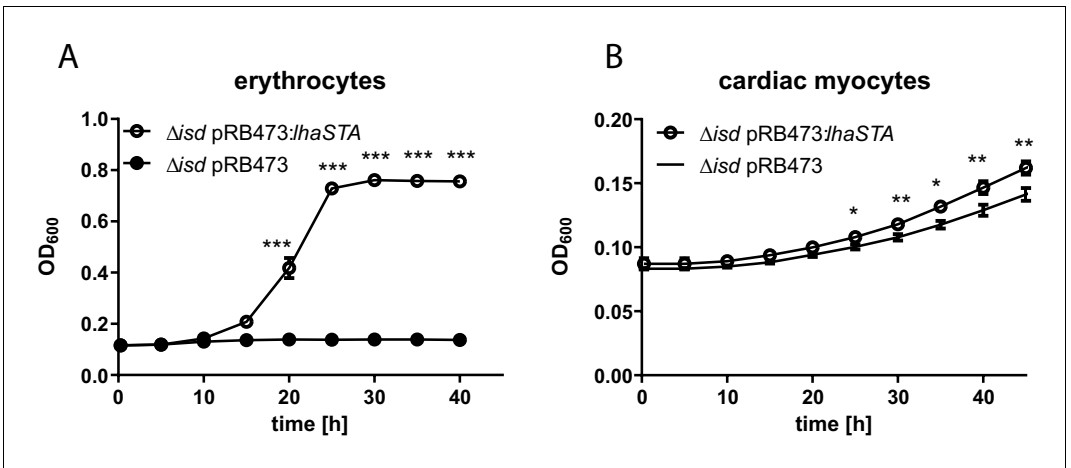

**Figure 5.** LhaSTA allows usage of host cells as an iron source. (**A**) Growth of *S. lugdunensis* N940135 Δ*isd* pRB473:*lhaSTA* and Δ*isd* pRB473 on human erythrocytes. Strains were grown in the presence of freshly isolated human erythrocytes ($10^5$ cells/ml) as a sole source of iron. 500 µl of cultures were inoculated to an $OD_{600}$ = 0,05 in 48 well plates and $OD_{600}$ was monitored every 15 min using an Epoch1 plate reader. For reasons of clarity values taken every 5 hr are displayed. Mean and SD of three experiments are shown. Statistical analysis was performed using students unpaired t-test. ***p<0,0001 (**B**) Growth of *S. lugdunensis* N940135 Δ*isd* pRB473 and Δ*isd* pRB473: *lhaSTA* on human cardiac myocytes. Strains were grown in the presence of 40000 primary human cardiac myocytes per well as a sole source of iron. Cardiac myocytes were detached and washed once with RPMI+200 µM EDDHA prior addition to the wells. 500 µl of cultures were inoculated to an $OD_{600}$ = 0,05 in 48 well plates and $OD_{600}$ was monitored every 15 min using an Epoch1 plate reader. For reasons of clarity values taken every 5 hr are displayed. Mean and SD of three experiments are shown. Statistical analysis was performed using students unpaired t-test. *p<0,05, **p<0,01.

The online version of this article includes the following figure supplement(s) for figure 5:

**Figure supplement 1.** Mouse systemic infection model.

**Figure supplement 2.** Hemolysis of human and mouse erythrocytes.

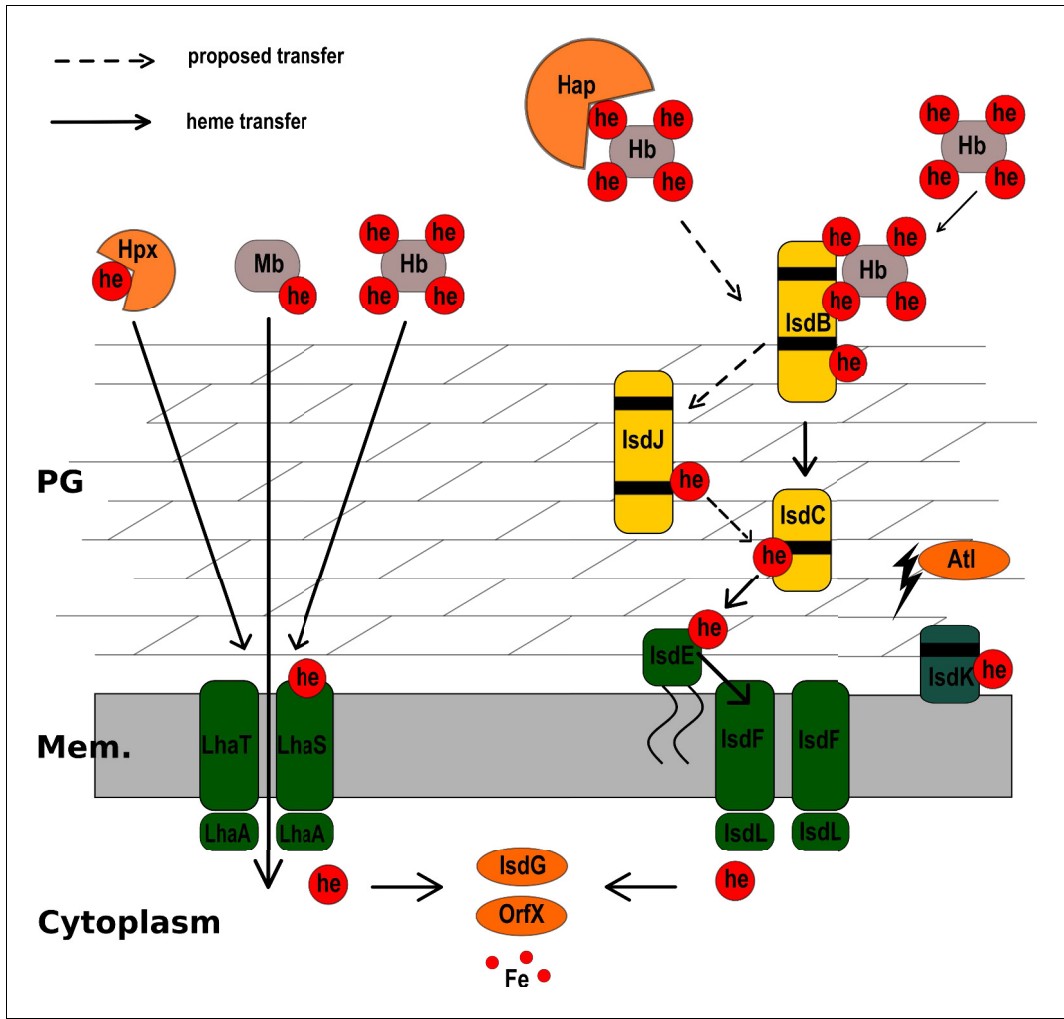

**Figure 6.** Model of heme acquisition in *S. lugdunensis*. ABC membrane transporters are shown in green. Cell wall-anchored proteins of the Isd-system are shown in yellow. Heme/hemoglobin-binding NEAT motifs within each protein are indicated as black boxes. Black arrows indicated the transfer of heme. he: heme; hb: hemoglobin; PG: peptidoglycan; Mem: Membrane; Hap: Haptoglobin; Hpx:Hemopexin; Atl: Autolysin.

*2002*). The usage of heme by invasive pathogens is widely distributed, however, the molecular pathways and hemoprotein substrate ranges differ dramatically (see *Choby and Skaar, 2016* for an excellent review). Iron dependent surface determinant loci are used to acquire heme from hemoglobin by several Gram-positive pathogens including *S. aureus* (*Mazmanian et al., 2003*), *S. lugdunensis* (*Heilbronner et al., 2011*; *Heilbronner et al., 2016*), *Bacillus anthracis* (*Skaar et al., 2006*), *Streptococcus pyogenes* (*Lei et al., 2002*) and *Listeria monocytogenes* (*Jin et al., 2006*).

ABC transporters of the Energy-coupling factor type (ECF) are trace element acquisition systems (*Finkenwirth and Eitinger, 2019*; *Rempel et al., 2019*). ECF-type transporters are characterized by high affinity towards their ligands and ECF systems specific for the vitamins riboflavin (*Duurkens et al., 2007*), folate (*Eudes et al., 2008*), thiamine (*Erkens and Slotboom, 2010*), biotin (*Berntsson et al., 2012*), cobalamine (*Santos et al., 2018*; *Rempel et al., 2018*), pantothenate (*Neubauer et al., 2009*; *Zhang et al., 2014*), niacin (*ter Beek et al., 2011*) and pyridoxamine (*Wang et al., 2015*) as well as for the trace metals nickel and cobalt (*Yu et al., 2014*; *Kirsch and Eitinger, 2014*) have been described. However, ECF-transporters that allow iron acquisition have so far remained elusive.

Now we show that *S. lugdunensis* encodes the iron regulated ECF-transporter LhaSTA. LhaS binds heme and enables accumulation of iron within the cytoplasm. Therefore, the system represents

a novel type of 'bona fide' iron acquisition system. Recombinant LhaS acquired heme from human and murine hemoglobin, from human and equine myoglobin as well as from human hemopexin. The ability of LhaS to accept heme from several sources strongly suggest an affinity-driven mechanism relying on passive diffusion of heme between proteins rather than on active extraction. Such a mechanism has been suggested for HasA-type hemophores of Gram-negative pathogens such as *Serratia marcescens*, *Yersina pestis* and *Pseudomonas aeruginosa* (*Wandersman and Delepelaire, 2012*; *Létoffé et al., 1999*). Similar to LhaS, HasA has been shown to possess a broad hemoprotein substrate range and allows the usage of hemoglobin from different species as well as myoglobin and hemopexin (*Wandersman and Delepelaire, 2012*). This ability of HasA was attributed to its high affinity towards heme (Kd = 0.2 nM) (*Deniau et al., 2003*). ECF-specificity subunits frequently possess Kds towards their ligands in the low nanomolar to picomolar range (*Rempel et al., 2019*), supporting the idea that LhaS might directly accept heme from hemoproteins. We attempted isothermal titration calorimetry to determine the affinity of LhaS towards heme, but our efforts failed to deliver a precise Kd. However, co-purification of heme with heterologously expressed LhaS suggests that the off-rates are low, consistent with high-affinity binding. Therefore, the system might be superior to heme acquisition systems, which depend on specific interactions between bacterial hemoprotein-receptors and host hemoproteins to extract heme. The *S. aureus* Isd system is well-studied in this regard. The surface located receptor IsdB binds hemoglobin through its N-terminal NEAT domain (IsdB-N1). This binding is proposed to induce a steric strain that facilitates heme dissociation. Heme is then captured by the C-terminal NEAT domain (isdB-N2) and transported across the cell wall and membrane (*Gianquinto et al., 2019*; *Sheldon and Heinrichs, 2015*; *Pilpa et al., 2009*; *Torres et al., 2006*; *Pishchany et al., 2014*). Similarly, the secreted hemophores IsdX1 and IsdX2 of *Bacillus anthracis* possess NEAT motifs and perform the same two-step process as IsdB of *S. aureus* to acquire heme (*Maresso et al., 2008*). This mechanism harbours the disadvantage of facilitating heme acquisition only from a single hemoprotein. IsdB allows acquisition from hemoglobin but does not interact with myoglobin or hemopexin (*Torres et al., 2006*) and even hemoglobin from different species reduces the efficacy of the system (*Choby et al., 2018*; *Pishchany et al., 2010*). The same is true for IsdB of *S. lugdunensis* (*Zapotoczna et al., 2012*). *Haemophilus influenza* uses the specific interaction between the surface exposed receptor HxuA and hemopexin to facilitate heme dissociation. Heme is subsequently captured by HxuC (*Zambolin et al., 2016*; *Hanson et al., 1992*). Again, the specificity for hemopexin prevents usage of other hemoproteins by HxuA. Specific interactions between LhaS and multiple host hemoproteins seem unlikely, suggesting that the superior affinity of LhaS towards the heme group bypasses the need for protein-protein interactions and enables usage of different hemoproteins. However, additional experimental evidence is required to strengthen this hypothesis of passive heme transfer.

The LhaSTA operon of *S. lugdunensis* is located within the *isd* operon which encodes the hemoglobin receptor IsdB, the cell wall-anchored, heme-binding proteins IsdJ and IsdC as well as the conventional heme membrane transporter IsdEFL. Deletion of *lhaSTA* in combination with *isdEFL* did not completely abrogate acquisition of free heme. A similar effect has been observed in *S. aureus* suggesting the presence of additional, low affinity heme transporters within these species (*Grigg et al., 2007*). Furthermore, a putative secreted/membrane associated hemophore (IsdK) is encoded within the operon (*Zapotoczna et al., 2012*). Interestingly, we show LhaSTA to be functionally independent of the Isd cluster because LhaSTA-dependent usage of all host hemoproteins except for Hb-Hap was observed in the absence of all Isd-associated proteins. This indicates that LhaSTA does not rely on Isd-dependent funneling of heme across the cell wall, but also raises interesting questions about the spatial organization of heme acquisition and donor proteins. For an efficient transfer of heme between host hemoproteins and LhaS one would expect that spatial proximity between the proteins is required. Yet, LhaS is situated in the bacterial membrane and host hemoproteins are too large (hemoglobin ~64–16 kd (tetramer-monomer), myoglobin ~16 kDa, Hemopexin-heme ~70,6 kDa) to readily penetrate the peptidoglycan layer of Gram-positive bacteria. However, it has been shown that staphylococcal peptidoglycan contains pores that might allow access of proteins to the bacterial membrane (*Kim et al., 2013*; *Turner et al., 2010*). Along this line, it is tempting to speculate that surface receptor-dependent acquisition of Hb-Hap might be needed as these complexes exceed 100 kDa and might be unable to access the bacterial membrane. However, we also observed that recombinant LhaS did not accept heme from Hb-Hap which might indicate that binding of haptoglobin to hemoglobin increases the strength of heme binding to the

protein complex. Such an effect of haptoglobin is to our knowledge not known and might be interesting for further investigation.

We failed to establish a functional mouse model of systemic disease to study the *in vivo* role of LhaSTA for the pathogenicity of *S. lugdunensis*. The reasons for this can be plentiful as little is known about virulence factors of *S. lugdunensis*. Genome analysis showed that *S. lugdunensis* lacks the wide variety of virulence and immune evasion molecules found in *S. aureus* (*Heilbronner et al., 2011*). This is the most likely explanation for the apparent reduced virulence of *S. lugdunensis* in mice. Nevertheless, the co-existence of the Isd and LhaSTA heme-acquisition system in this species may represent a virulent trait and be required for invasive disease. In line with this, most *S. lugdunensis* strains are highly hemolytic on blood agar plates suggesting that the release of hemoproteins from host cells can be achieved by this species. The hemolytic SLUSH peptides (*Donvito et al., 1997*) of *S. lugdunensis* resemble the β-PSMs of *S. aureus* (*Rautenberg et al., 2011*). Additionally, the sphingomyelinase C (ß-toxin) is conserved in *S. lugdunensis* (*Heilbronner et al., 2011*). However, recent research suggested that S. *aureus* targets erythrocytes specifically using the bi-component toxins LukED and HlgAB recognising the DARC receptor (*Spaan et al., 2015*). This creates human specificity. Whether similar mechanisms are used by *S. lugdunensis* is unclear, but we found that, in contrast to human cells, *S. lugdunensis* failed to lyse murine erythrocytes. This suggests that host specific virulence factors are present in *S. lugdunensis*. Bi-component toxin genes are not located in the chromosome but genes encoding a streptolysin-like toxin were identified (*Heilbronner et al., 2011*).

We found that LhaSTA facilitated growth of *S. lugdunensis* in the presence of human cells such as erythrocytes and cardiac myocytes strongly suggesting that the system allows usage of these cells during invasive disease.

Altogether our experiments identify LhaSTA as an ECF-transporter able to acquire iron and place this important class of nutrient acquisition system in the context of bacterial pathogenesis and immune evasion strategies. During the revision of this manuscript Chatterjee and colleagues published the identification of a heme-specific ECF transporter in streptococci (*Chatterjee et al., 2020*). In addition, a preprint manuscript that reports the identification of a heme-specific ECF transporter in *Lactococcus sakei* is present in bioarchives (*Verplaetse, 2019*). Although these transporters seem functionally redundant to the one of *S. lugdunensis* described here, the specificity subunits of the systems show remarkably little amino-acid sequence similarity. This suggests that the genes encoding them might have developed independently in bacterial species. This was also suggested for the cobalamin-specific components BtuM and CbrT which bind the same ligand despite little sequence conservation (*Santos et al., 2018*).

Additional experiments are required to determine whether heme-specific ECF transporters are also present in other bacterial pathogens and the biochemical properties of heme-binding need to be further characterized to better understand the role of these systems in overcoming nutritional iron limitation.

# Materials and methods

## Key resources table

| Reagent type (species) or resource | Designation | Source or reference | Identifiers | Additional information |
|---|---|---|---|---|
| Strain, strain background (*Staphylococus lugdunensis*) | N940135 | National Reference Center for Staphylococci, Lyon, France (*Heilbronner et al., 2011*) | | |
| Strain, strain background (*S. lugdunensis*) | N920143 | National Reference Center for Staphylococci, Lyon, France (*Heilbronner et al., 2011*) | | |
| Strain, strain background (*S. lugdunensis*) | N920143 Δ*isdEFL* | This paper | | Markerless deletion mutant of *isdEFL* |

*Continued on next page*

*Continued*

| Reagent type (species) or resource | Designation | Source or reference | Identifiers | Additional information |
|---|---|---|---|---|
| Strain, strain background (*S. lugdunensis*) | N920143 Δ*lhaSTA* | This paper | | Markerless deletion mutant of *lhaSTA* |
| Strain, strain background (*S. lugdunensis*) | N920143 Δ*isdEFL*Δ*lhaSTA* | This paper | | Markerless double deletion mutant of *isdEFL* and *lhaSTA* |
| Cell line (Human) | Human cardiac myocytes (HCM) | PromoCell | C-12810 | |
| Recombinant DNA reagent | pQE-30 | Qiagen | | IPTG inducible expression plasmid |
| Recombinant DNA reagent | pQE30:lhaS | This paper | | LhaS expressing plasmid for protein purification |
| Recombinant DNA reagent | pRB473: lhaSTA | This paper | | LhaSTA expressing plasmid for complementation |
| Recombinant DNA reagent | pIMAY (plasmid) | *Monk et al., 2012* | See Material and methods | Thermosensitive vector for allelic exchange |
| Recombinant DNA reagent | pIMAY:Δ*isd* | *Zapotoczna et al., 2012* | | Plasmid for the deletion of the entire *isd* locus |
| Recombinant DNA reagent | pIMAY:Δ*isdEFL* | This study | | Plasmid for the deletion of conventional heme transporter *isdEFL* |
| Recombinant DNA reagent | pIMAY:Δ*lhaSTA* | This study | | Plasmid for the deletion of heme specific ECF-transporter |
| Recombinant DNA reagent | pRB473 | *Brückner, 1992* | | Expression plasmid without promotor region. |
| Biological sample (Human) | Human hemoglobin | Own preparation | See Material and methods | Sex male |
| Biological sample (Pork) | Porcine hemin | Sigma | 51280 | |
| Biological sample (Human) | Human Myoglobin | Sigma Aldrich | M6036 | |
| Biological sample (Horse) | Equine Myoglobin | Sigma Alrich | M1882 | |
| Biological sample (Human) | Human Haptoglobin (Phenotype 1–1) | Sigma Aldrich | SRP6507 | |
| Biological sample (Human) | Human Hemopexin | Sigma Aldrich | H9291 | |
| Chemical compound, drug | RPMI 1640 Medium | Sigma Aldrich | R6504-10L | |
| Chemical compound, drug | Casamino acids | BACTO | 223050 | |
| Chemical compound, drug | EDDHA | LGC Standarts | TRC-E335100-10MG | |
| Chemical compound, drug | Dodecyl-β-D-maltosid (DDM) | Carl Roth | CN26.1 | |
| Chemical compound, drug | 3,3′,5,5′-tetramethylbenzidine (TMBZ) | Sigma Aldrich | 860336 | |
| Chemical compound, drug | Profinity IMAC resin nickel chrged | BIO RAD | 1560135 | |

## Chemicals

If not stated otherwise, reagents were purchased from Sigma.

## Bacterial strains and growth in iron limited media

All bacterial strains generated and/or used in this study are listed in Key resources table. For growth in iron limited conditions, bacteria were grown overnight in Tryptic Soy Broth (TSB) (Oxoid). Cells were harvested by centrifugation, washed with RPMI containing 10 µM EDDHA (LGC standards), adjusted to an $OD_{600}$ = 1 and 2,5 µl were used to inoculate 0,5 ml of RPMI+ 1% casamino acids (BACTO) + 10 µM EDDHA in individual wells of a 48 well microtiter plate (NUNC). As sole iron source 200 nM porcine hemin (Sigma), 2.5 µg/ml human hemoglobin (own preparation), 10 µg/ml human myoglobin (Sigma) or equine myoglobin (Sigma), 117 nM human haptoglobin-hemoglobin or 200 nM hemopexin-heme (Sigma) were added to the wells. Bacterial growth was monitored using an Epoch2 reader (300 rpm, 37°C). The $OD_{600}$ was measured every 15 min.

## Creation of markerless deletion mutants in *S. lugdunensis*

For targeted deletion of *lhaSTA* and *isdEFL*, 500 bp DNA fragments upstream and downstream of the genes to be deleted were amplified by PCR. A sequence overlap was integrated into the fragments to allow fusion and creating an ATG-TAA scar in the mutant allele. The 1 kb deletion fragments were created using spliced extension overlap PCR and cloned into pIMAY. All the oligonucleotides are summarized in *Supplementary file 1* Targeted mutagenesis of *S. lugdunensis* was performed using allelic exchange described elsewhere (*Monk et al., 2012*). The plasmids and the primers used are listed in key resources table and *Supplementary file 1*, respectively.

## Heterologous expression of LhaS and membrane vesicle preparation

LhaS was overexpressed with a N-terminal deca-His tag using pQE-30 in *E. coli* XL1 blue in either Lysogeny broth (LB) medium or RPMI+1% casamino acids. 100 ml overnight culture in LB with 100 µg ml$^{-1}$ ampicillin was harvested by centrifugation and washed once in PBS. Cells were resuspended in 5 ml PBS and used for inoculation of 2 L RPMI + 1% casamino acids or LB medium. Cells were allowed to grow at 37°C to an $OD_{600}$ = 0.6–0.8. Expression was induced by adding 0.3 mM IPTG for 4–5 hr at 25°C. Cells were harvested, washed with 50 mM potassium phosphate buffer (KPi) pH 7.5, and lysed through 3 rounds of sonication (Branson Digital Sonifier; 2 min, 30% amplitude), in presence of 200 µM PMSF, 1 mM $MgSO_4$ and DNaseI. Cell debris were removed by centrifugation for 30 min at 7000 rpm and 4°C. The supernatant was centrifuged for 2 hr at 35000 rpm and 4°C to collect membrane vesicles (MVs). The MV pellet was homogenized in 50 mM KPi pH 7.5 and flash frozen in liquid nitrogen, stored at −80°C and used for purification.

## Purification of LhaS

His-tagged LhaS MVs were dissolved in solubilisationbuffer (50 mM KPi pH 7.5, 200 mM KCl, 200 mM NaCl, 1% (w/v) n-dodecyl-b-D-maltopyranosid (DDM, Roth) for 1 hr at 4°C on a rocking table. Non-soluble material was removed by centrifugation at 35000 rpm for 30 min and 4°C. The supernatant was decanted into a poly-prep column (BioRad) containing a 0.5 ml bed volume $Ni^{2+}$-NTA sepharose slurry, equilibrated with 20 column volumes (CV) wash buffer (50 mM KPi pH 7.5, 200 mM NaCl, 50 mM imidazole, 0.04% DDM) and incubated for 1 hr at 4°C while gently agitating. The lysate was drained out of the column and the column was washed with 4 CV wash buffer. Bound protein was eluted from the column in three fractions with elution buffer (50 mM KPi, pH 7.5, 200 mM NaCl, 350 mM imidazole, 0.04% DDM). The sample was centrifuged for 3 min at 10.000 rpm to remove aggregates and loaded on a Superdex 200 Increase 10/300 GL gel filtration column (GE Healthcare), which was equilibrated with SEC buffer (50 mM KPi pH 7.5, 200 mM NaCl, 0.04% DDM). Peak fractions were combined and concentrated in a Vivaspin disposable ultrafiltration device (Sartorius Stedim Biotec SA).

## MV saturation with hemoproteins

MVs (120 mg total protein content) from RPMI were thawed and incubated for 10 min at RT with each of the following molecules: 5.6 µM heme, 476 µg/ml human hemoglobin, 437 µg/ml equine myoglobin, 5.6 µM hemopexin-heme, 476 µg/ml hemoglobin-haptoglobin. Further purification was performed as described above. After $Ni^{2+}$ affinity chromatography the protein was concentrated and used to measure the peroxidase activity of heme (TMBZ staining).

## TMBZ staining of heme

Protein content was determined by Bradford analysis (BIORAD) according to the manufacturer's protocol. 15 µg protein sample was mixed 1:1 with native sample buffer (BIORAD) and loaded on a Mini-PROTEAN TGX Precast Gel (BIORAD). The PAGE was run at 4°C and low voltage for 2 hr in Tris/Glycine buffer (BIORAD). The gel was rinsed with $H_2O$ for 5 min and stained with 50 ml staining solution (15 ml 3,3',5,5'-tetramethylbenzidine (TMBZ) solution (6.3 mM TMBZ in methanol) +35 ml 0.25 M sodium acetate solution (pH 5)) for 1 hr at room temperature (RT) while gently agitating. The gel was then incubated for 30 min at RT in the dark in presence of 30 mM $H_2O_2$. The background staining was removed by incubating the gel in a solution of isopropanol/0.25 M sodium acetate (3:7). Following scanning, the gel was completely destained in a solution of isopropanol/0.25 M sodium acetate (3:7) and stained with the BlueSafe stain (nzytech) for 10 min.

## Preparation of human erythrocytes

Human blood was obtained from healthy volunteers and mixed 1:1 with MACS buffer (PBS w/o + 0.05% BSA + 2 mm EDTA). Erythrocytes were pelleted by density gradient centrifugation in a histopaque blood gradient for 20 min 380 x g at RT. The erythrocyte pellet was washed three times with erythrocyte wash buffer (21 mM Tris, 4.7 mM KCl, 2 mM $CaCl_2$, 140.5 mM NaCl, 1.2 mM $MgSO_4$, 5.5 mM Glucose, 0.5% BSA, pH 7.4). Cell count and viability was determined by using the trypan blue stain (BIO RAD).

## Purification of human hemoglobin

Human/murine haemoglobin was purified by using standard procedures describe in detail elsewhere (*Pishchany et al., 2013*).

## Preparation of saturated hemopexin and haptoglobin

Human hemopexin was dissolved in sterile PBS and saturated with porcine heme in a hemopexin: heme 1: 1.3 molar ratio for 1 hr at 37°C. This was followed by 48 hr dialysis in a Slide-a-Lyzer chamber (ThermoFisher) with one buffer (1 x PBS) change. Haptoglobin was saturated by mixing 4.7 µg/ml haemoglobin with 8.4 µg/ml human haptoglobin for 30 min at 37°C.

## Quantification of intracellular iron

Bacteria were grown at 37°C in RPMI + 1% casamino acids to an $OD_{600}$ = 0.6. An aliquot of the culture was collected prior addition of 5 µM heme and 25 µM EDDHA and further incubation at 37°C for 3 hr. At this time point bacteria were collected and resuspended in buffer WB (10 mM Tris-HCl, pH 7, 10 mM $MgCl_2$, 500 mM sucrose) to an $OD_{600}$ = 50. The bacterial pellet was collected by centrifugation at 8000 rpm for 7 min and resuspended in 1 ml buffer DB (10 mM Tris-HCl, pH 7, 10 mM $MgCl_2$, 500 mM sucrose, 0.6 mg/ml lysostaphin, 25 U/ml mutanolysin, 30 µl protease inhibitor cocktail (one complete mini tablet dissolved in 1 ml $H_2O$ (Roche), 1 mM phenyl-methanesulfonylfluoride (Roth). The cell wall was digested by incubating at 37°C for 1.5 hr, followed by centrifugation at 17000 x g for 10 min at 4°C. Pelleted protoplasts were washed with 1 ml buffer WB, centrifuged and resuspended in 200 µl buffer LB (100 mM Tris-HCl; pH 7, 10 mM $MgCl_2$, 100 mM NaCl, 100 µg/ml DNaseI, 1 mg/ml RNaseA). Protoplast lysis was performed through repeated cycles (3) of freezing and thawing. The lysate was centrifuged 30 min to pellet membrane fraction and recover the supernatant, which contained the cytosolic fraction and was used for quantification of total intracellular iron content.

Quantification of intracellular iron content by heme uptake was carried out according to *Riemer et al., 2004* with minor modifications. Briefly, 100 µl of the cytosolic fraction were mixed with 100 µl 50 mM NaOH, 100 µl HCL, and 100 µl iron releasing reagent (1:1 freshly mixed solution of 1.4 M HCl and 4.5% (w/v) $KMnO_4$ in $H_2O$). Samples were incubated for 2 hr at 60°C in a fume hood. 30 µl iron detection reagent (6.5 mM ferrozine, 6.5 mM neocuproine, 2.5 M ammonium acetate, 1 M ascorbic acid) was mixed with the samples and incubated for 30 min at 37°C while shaking (1100 rpm). Samples were centrifuged for 3 min at 12000 x g to remove precipitates. 150 µl of the supernatants were transferred to a 96-well microtiter plate and absorbance at 550 nm was measured in a plate reader (BMG Labtech). For determination of iron concentration, $FeCl_3$ standards in a range of 0 to 100 µM were prepared.

## Measurement of LhaS absorption spectra

LhaS was purified from LB (holo LhaS) or RPMI (apo LhaS) as described above. 2 µl protein sample were loaded on an Eppendorf µCuvette and absorptions spectra were measured at 260–620 nm with a BioPhotometer (Eppendorf).

## Characterization of LhaS and heme by mass spectrometry analysis

MALDI-TOF mass spectra were recorded with a Reflex IV (Bruker Daltonics, Bremen, Germany) in reflector mode. Positive ions were detected and all spectra represent the sum of 50 shots. A peptide standard (Peptide Calibration Standard II, Bruker Daltonics) was used for external calibration. 2,5-dihydroxybenzoic acid (DHB, Bruker Daltonics) dissolved in water/acetonitrile/trifluoroacetic acid (50/49.05/0.05) at a concentration of 10 mg ml$^{-1}$ was used as matrix. Before the measurements, the samples Lhas-apo (317 µg ml$^{-1}$) and Lhas+heme (377 µg ml$^{-1}$) were centrifuged and diluted with MilliQ-$H_2O$ (1:25). An aliquot of 1 µL of the samples was mixed with 1 µL of the matrix and spotted onto the MALDI polished steel sample plate. As the solution dried, the organic solvent evaporated quickly. At this point, the remaining mini droplet was removed gently with a pipette and the remaining sample was air-dried at room temperature.

High resolution mass spectra of Lhas-apo (317 µg ml$^{-1}$) and Lhas+heme (377 µg ml$^{-1}$) were recorded on a HPLC-UV-HR mass spectrometer (MaXis4G with Performance Upgrade kit with ESI-Interface, Bruker Daltonics). The samples were diluted with MilliQ-$H_2O$ (1:25) and 3 µL were applied to a Dionex Ultimate 3000 HPLC system (Thermo Fisher Scientific), coupled to the MaXis 4G ESI-QTOF mass spectrometer (Bruker Daltonics). The ESI source was operated at a nebulizer pressure of 2.0 bar, and dry gas was set to 8.0 L min$^{-1}$ at 200˚C. MS/MS spectra were recorded in auto MS/MS mode with collision energy stepping enabled. Sodium formate was used as internal calibrant. The gradient was 90% MilliQ-$H_2O$ with 0.1% formic acid and 10% methanol with 0.06% formic acid to 100% methanol with 0.06% formic acid in 20 min with a flow rate of 0.3 mL/min on a NucleoshellEC RP-$C_{18}$ (150 × 2 mm, 2.7 µm) from Macherey-Nagel.

[M+H]$^+$ calculated for $C_{34}H_{32}FeN_4O_4^+$: 616.1767; found 616.1778 (Δ ppm 1.78).

## Calculation of binding stoichiometry

To calculate the putative binding stoichiometry of heme and LhaS the heme concentration in *Figure 2A* was determined utilizing the extinction coefficient of 58,4 mM-$^1$ cm$^{-1}$ at 384 nm for heme. The LhaS concentration was determined utilizing the extinction coefficient of 29910 M-1 cm-1 (calculated with ProtParam tool – ExPASy) at 280 nm.

## Human Cardiac Myocytes (HCM)

Primary human cardiac myocytes were purchased from PromoCell (the identity of the cell line was not verified; the culture was negative for mycoplasma) and in 75-cm$^2$ culture flasks in 20 ml of myocyte growth medium (PromoCell). Cells were detached with accutase, washed once with RPMI containing 200 µM EDDHA and resuspended in PRMI containing 200 µM EDDHA. 40000 cells per well were used for bacterial growth assays as described above.

## Assessing hemolytic activity of *S. lugdunensis* culture supernatants

*S. aureus* and *S. lugdunensis* were grown overnight in TSB. Cells were pelleted and culture supernatants were filter sterilized using a 0.22 µM filter. A 100 µL volume of supernatant was added into 1 mL of PBS containing either 5% v/v murine or human red blood cells. Mixtures were incubated at room temperature without shaking for 48 hr.

## Acknowledgements

We thank Timothy J Foster and Libera Lo Presti for critically reading and editing this manuscript. We thank Andreas Peschel for helpful discussion. We thank Sarah Rothfuß and Vera Augsburger for excellent technical support and Imran Malik for the introduction to the ITC technology.

## Additional information

### Funding

| Funder | Grant reference number | Author |
|---|---|---|
| Deutsche Forschungsge-meinschaft | HE8381/3-1 | Simon Heilbronner |
| Deutsche Forschungsge-meinschaft | EXC2124 | Simon Heilbronner |
| Nederlandse Organisatie voor Wetenschappelijk Onderzoek | TOP grant 714.018.003 | Dirk J Slotboom |
| Canadian Institutes of Health Research | PJT-153308 | David E Heinrichs |
| Deutsche Forschungsge-meinschaft | GRK 1708 | Simon Heilbronner |

The funders had no role in study design, data collection and interpretation, or the decision to submit the work for publication.

### Author contributions

Angelika Jochim, Data curation, Investigation, Visualization, Methodology, Writing - review and editing; Lea Adolf, Data curation, Investigation, Methodology, Writing - review and editing; Darya Beli-kova, Nadine Anna Schilling, Data curation, Investigation, Methodology; Inda Setyawati, Denny Chin, Severien Meyers, Data curation, Methodology; Peter Verhamme, David E Heinrichs, Dirk J Slot-boom, Supervision, Writing - review and editing; Simon Heilbronner, Conceptualization, Funding acquisition, Methodology, Writing - original draft

### Author ORCIDs

Severien Meyers https://orcid.org/0000-0003-3481-4793
David E Heinrichs http://orcid.org/0000-0002-7217-2456
Dirk J Slotboom http://orcid.org/0000-0002-5804-9689
Simon Heilbronner https://orcid.org/0000-0002-6774-2311

### Ethics

Human subjects: Human Erythrocytes were isolated from venous blood of healthy volunteers in accordance with protocols approved by the Institutional Review Board for Human Subjects at the University of Tübingen. Informed written consent was obtained from all volunteers.
Animal experimentation: Animal experiments were performed in strict accordance with the European Health Law of the Federation of Laboratory Animal Science Associations. The protocol was approved by the Regierungspräsidium Tübingen (IMIT1/17).

### Decision letter and Author response

Decision letter https://doi.org/10.7554/eLife.57322.sa1
Author response https://doi.org/10.7554/eLife.57322.sa2

## Additional files

### Supplementary files

• Supplementary file 1. Key Resources Table PCR primers. PCR and qPCR primers sequences used in this study.

• Transparent reporting form

## Data availability

The datasets gained during the current study are available at Dryad Digital Repository: https://doi.org/10.5061/dryad.fqz612jqc.

The following dataset was generated:

| Author(s) | Year | Dataset title | Dataset URL | Database and Identifier |
|---|---|---|---|---|
| Jochim A, Adolf LA, Belikova D, Schilling NA, Setyawati I, Chin D, Meyers S, Verhamme P, Heinrichs DE, Slotboom DJ, Heilbronner S | 2020 | Data from: An ECF-type transporter scavenges heme to overcome iron-limitation in Staphylococcus lugdunensis | https://doi.org/10.5061/dryad.fqz612jqc | Dryad Digital Repository, 10.5061/dryad.fqz612jqc |

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
