## [Decision Letter]

**Acceptance summary:**

You present here a strong piece of work describing the identification and initial characterization of a heme ECF transporter (LhaSTA) in the opportunistic human pathogen *Staphylococcus lugdunensis* that could allow the scavenging of iron from host cells. The manuscript is well written and the study is an important contribution since it reveals that heme is a substrate of ECF transporters.

**Decision letter after peer review:**

Thank you for submitting your article "An ECF-type transporter scavenges heme to overcome iron-limitation in *Staphylococcus lugdunensis*" for consideration by *eLife*. Your article has been reviewed by two peer reviewers, and the evaluation has been overseen by a Reviewing Editor and Jos van der Meer as the Senior Editor. The reviewers have opted to remain anonymous.

The reviewers have discussed the reviews with one another and the Reviewing Editor has drafted this decision to help you prepare a revised submission.

You present here a strong piece of work describing the identification and initial characterization of a heme ECF transporter (LhaSTA) in the opportunistic human pathogen *Staphylococcus lugdunensis* that could allow the scavenging of iron from host cells. The manuscript is well written and the study is an important contribution since it reveals that heme is a substrate of ECF transporters.

To improve the clarity and presentation of the work, authors are asked to provide the following modifications to their manuscript:

1) Subsection “LhaS binds heme”, last paragraph and Supplementary Figure 2: The outcome of the ITC experiments is unsatisfactory. According to Figure 4D, "instability" of LhaS in detergent solution in the absence of heme is not obvious on a proteolytic scale. It is not uncommon that a membrane protein's function depends on its lipid environment and is significantly impaired in detergent solution. There are several ways to overcome this problem. The authors could have considered to reconstitute LhaS into proteoliposomes or lipid nanodiscs prior to measurements. Potential alternatives include extraction of the protein and its lipid environment using styrene/maleic acid or diisobutylene/maleic acid copolymers without using detergents. However, the reviewers agreed that the key message of the paper is fully substantiated by the rest of the manuscript without having to incorporate this result. Please remove from the manuscript.

2) Figure 1D: Could the authors discuss what allows growth on heme on Figure 1D? How is heme transported in the isdEF/lhaSTA double mutant?

3) "LB medium, which is known to contain heme". Could the authors provide a reference?

4) The authors are probably aware of a preprint posted on December 6, 2019 on a heme ECF transporter in *Lactobacillus sakei* (Verplaetse et al., 2019). Please consider citing it.

5) Abstract: Although the list of experimentally verified or predicted substrates of ECF transporters is not very long, "only a handful" is an understatement.

6) Subsection “Background”, first paragraph: iron, manganese, copper and zinc are mentioned, but cobalt and nickel (for uptake of which ECF transporters exist) are not.

---

## [Author Response]

[…] To improve the clarity and presentation of the work, authors are asked to provide the following modifications to their manuscript:1) Subsection “LhaS binds heme”, last paragraph and Supplementary Figure 2: The outcome of the ITC experiments is unsatisfactory. According to Figure 4D, "instability" of LhaS in detergent solution in the absence of heme is not obvious on a proteolytic scale. It is not uncommon that a membrane protein's function depends on its lipid environment and is significantly impaired in detergent solution. There are several ways to overcome this problem. The authors could have considered to reconstitute LhaS into proteoliposomes or lipid nanodiscs prior to measurements. Potential alternatives include extraction of the protein and its lipid environment using styrene/maleic acid or diisobutylene/maleic acid copolymers without using detergents. However, the reviewers agreed that the key message of the paper is fully substantiated by the rest of the manuscript without having to incorporate this result. Please remove from the manuscript.

We agree that the results of the ITC experiments are of little additional value to this manuscript. Hence, we removed Supplementary Figure 2 as well as the description of the experiment from the Results and the Materials and methods sections. However, we still mention in the Discussion section that we attempted ITC but failed at this stage (third paragraph). We thank the reviewer for suggesting alternative experimental approaches which we will consider for future experiments.

2) Figure 1D: Could the authors discuss what allows growth on heme on Figure 1D? How is heme transported in the isdEF/lhaSTA double mutant?

These results mimic observations made in *Staphylococcusaureus*. In this staphylococcal species heme membrane transport has been extensively studied and bacterial growth on heme as sole iron source was shown to be only mildly affected by isdE deletion [1, 2]. While the precise mechanism remains elusive, it was speculated that a so far unidentified (low affinity) heme transport system allows acquisition of free heme in vitro, but is of little relevance when heme is bound to hemoproteins. Such a system might be conserved in *S. lugdunensis*. This is now added to the text (Discussion, fourth paragraph). Interestingly, we observed that growth using heme is reduced strongest when the entire operon of *S. lugdunensis* (isdB to atlI) is deleted. This suggests that other proteins within the locus (in addition to LhaSTA and IsdEF) do also contribute to transport of heme over the membrane.

We suspect that IsdK might be involved, which is a membrane-located NEAT domain-containing and thereby heme-binding protein. This is an unusual protein [3]. This is currently investigated in our laboratory and therefore not discussed in detail in this manuscript.

3) "LB medium, which is known to contain heme". Could the authors provide a reference?

The yeast extract present in LB contains significant amounts of heme which must be the source of heme in our assays [4]. The reference is now provided (subsection “LhaS binds heme”).

4) The authors are probably aware of a preprint posted on December 6, 2019 on a heme ECF transporter in Lactobacillus sakei (Verplaetse et al., 2019). Please consider citing it.

Yes, we are aware of the preprint. In addition, we note that a heme-ECF transporter was very recently identified in streptococcal species. Interestingly, both the *Lactobacillus* and the Streptococcus systems show little similarity to the Lha of *S. lugdunensis*. A comparison of the three systems is now incorporated in the Discussion section.

5) Abstract: Although the list of experimentally verified or predicted substrates of ECF transporters is not very long, "only a handful" is an understatement.

This is now changed to “a number of molecules…”

6) Subsection “Background”, first paragraph: iron, manganese, copper and zinc are mentioned, but cobalt and nickel (for uptake of which ECF transporters exist) are not.

Cobalt and nickel have been added (subsection “Background”, first paragraph) and references for the respective ECF-systems are given in the Discussion (third paragraph).

References:

1) Grigg, J.C., et al., Heme coordination by *Staphylococcus aureus* IsdE. J Biol Chem, 2007. 282(39): p. 28815-22.

2) Sheldon, J.R. and D.E. Heinrichs, Recent developments in understanding the iron acquisition strategies of gram positive pathogens. FEMS Microbiol Rev, 2015. 39(4): p. 592-630.

3) Zapotoczna, M., et al., Iron-Regulated Surface Determinant (Isd) Proteins of *Staphylococcus lugdunensis*. J Bacteriol, 2012. 194(23): p. 6453-67.

4) Fyrestam, J. and C. Ostman, Determination of heme in microorganisms using HPLC-MS/MS and cobalt(III) protoporphyrin IX inhibition of heme acquisition in *Escherichia coli*. Anal Bioanal Chem, 2017. 409(30): p. 6999-7010.